# Adjusting for truncated study duration in recurrent event analysis: A weighting approach for clinical trials

John Michael Raj A[1,2], Tinku Thomas[2]*, Pratibha Dwarkanath[3]

1 Center for Doctoral Studies, Manipal Academy of Higher Education, Manipal, India, 2 Department of Biostatistics, St. John's Medical college, Bangalore, India, 3 Division of Nutrition, St. John's Research Institute, Bangalore, India

* tinku.sarah@sjri.res.in

## Abstract

### Background

In recurrent event analysis with fixed follow-up intervals, truncated follow-up due to early dropout or study termination introduces bias and reduces precision in risk estimates, particularly in clinical trials where shorter observation periods may underestimate event risks.

### Methods

We propose a time-based weighting approach using the ratio of observed-to-expected follow-up duration in the Prentice-Williams-Peterson Gap Time (PWP-GT) model. The method was evaluated in simulations and applied to a double-blinded trial (N = 4000) comparing 500 mg vs. 1500 mg daily calcium supplementation for preeclampsia prevention. For demonstration of the problem and application of the weighting method, drug non-adherence at follow-up visits was considered as the recurrent event.

### Results

Simulations showed the weighted PWP-GT model had lower bias (1.0% vs. 1.3%) and improved precision compared to the unweighted model, with coverage probabilities >94%. In the trial data, weighting yielded smaller standard errors and a more conservative hazard ratio for hypertension family history (weighted HR = 1.14, SE = 0.054 vs. unweighted HR = 1.23, SE = 0.065).

### Conclusion

Unaccounted truncated follow-up in recurrent event studies can bias the risk estimation if unaccounted for. Our findings demonstrate that total time-based weighting

**Data availability statement:** The dataset used for demonstration in this manuscript is derived from a third-party randomized controlled trial: Dwarkanath P, Muhihi A, Sudfeld CR, Wylie BJ, Wang M, Perumal N, et al. Two Randomized Trials of Low-Dose Calcium Supplementation in Pregnancy. N Engl J Med. 2024;390(2):143–53. The authors do not own the rights to share these data; all data-sharing rights remain exclusively with the study Principal Investigator (PI). As detailed in the original publication's data-sharing policy, complete de-identified patient-level data from the trial may be made available to qualified researchers. Access is granted only upon formal request to the PI, accompanied by a research proposal and the necessary approvals from the Institutional Ethics Committee (IEC) and other relevant regulatory authorities. The PI details, as provided in the original trial's data-sharing statement, are: Dr. Wafaie Fawzi Harvard T.H. Chan School of Public Health 665 Huntington Avenue Boston, MA 02115 Email: mina@hsph.harvard.edu.

**Funding:** The author(s) received no specific funding for this work.

**Competing interests:** The authors have declared that no competing interests exist.

**Abbreviations:** PWP-GT: Prentice, William, and Peterson Gap time model; PWP-CP: Prentice, William, and Peterson counting process model; IPTW: Inverse probability of treatment weighting; IPCW: Inverse probability of censoring weighting; AG-CP: Anderson and Gill Counting process model.

effectively addresses this bias and enhances precision in both simulated and real datasets.

## Introduction

Time-to-first event data are commonly analyzed using Cox proportional hazard model. However, recurrent events challenges the Cox model assumptions due to correlation between events or event dependency within individuals [1]. Modifications of the Cox model such as extended Cox models or variance corrected models are effective in handling event dependency [2–4].

In addition to the dependency of events, recurrent event analysis also faces issues with informative censoring, confounding bias, covariate imbalance, selection bias, early dropouts and competing risk, as in regular survival analysis models. These complexities in the data can lead to biased and less-precise estimates in the Cox regression model, which have been addressed using a variety of methods, including marginal, conditional, and joint models [5,6]. Weighted estimators have been suggested to mitigate bias and imprecision in estimation arising from these complexities. Application of weights in time to event analysis was first introduced by Prentice (1986) [7] to adjust for the exposure imbalance in the case cohort design. Later, Inverse probability weights (IPW) were effectively applied to adjust for biases induced by informative censoring, dropouts rate imbalances and treatment imbalances due to confounders. The inverse probability censoring weight (IPCW) were used to account for covariate dependent censoring and competing event induced censoring, whereas inverse probability treatment weight (IPTW) was used to address covariate dependent treatment allocation and time dependent covariates in both time to first event and time to recurrent event analysis [8–13].

Although several methods have been proposed to address different biases, one overlooked issue is the impact of truncated follow-up due to dropout or early termination of participation, especially in trials involving measurement of recurrent events at fixed follow up interval. This introduces significant bias in risk estimation for recurrent events [14]. The risk of recurrent events is restricted for individuals with shorter follow up durations due to truncated follow up. For instance, in a clinical trial among pregnant women studying the effect of low-dose calcium supplementation on preeclampsia incidence, adherence was assessed at monthly follow-ups until delivery. Participants with shorter study durations had lower observed nonadherence rates, not necessarily due to better adherence but because of reduced overall exposure time to calcium supplementation or truncated follow up in some individuals. This artificially lower event rate due to early dropout or fewer follow-up visits induces bias and imprecision in estimated hazard ratios for recurrent events. To ensure accurate risk estimation, the model must account for this truncation.

This study examined whether individual participant weights generated from the ratio of actual to expected follow up time duration, can reduce bias and improve precision of estimates in studies with truncated follow-up duration due to early discontinuation. This was assessed using simulated and real time clinical trial secondary data.

## Methods

### Statistical analysis

**Model.** For our recurrent event analysis, we chose the Prentice, William, and Peterson Gap time (PWP-GT) model [15]. The PWP-GT model is a conditional model that assumes that recurrent events within the participant are related and that a participant is not at risk for the $k^{th}$ event until he/she experiences the $(k-1)^{th}$ event. PWP-GT model is given by:

$$h_{ik}(t \mid \boldsymbol{X}) = h_{0k}(t - t_{k-1})e^{X_{ik}\beta}$$

Where $t - t_{k-1}$ is the gap time for the given event.

**Simulation.** The main objective of the simulation was to examine the unbiasedness of the estimates with and without sampling weights in simulated recurrent event data in a fixed follow-up duration design. In general, the recurrent event data are simulated based on standard recurrent event scenarios where every study participant will have multiple visits where visits are separated by recurrent events (i.e., there is no fixed follow-up interval to assess the event of interest). In this design censoring can occur only in the last visit due to study completion. In contrast, in recurrent event studies with fixed follow-up duration, the event is recorded at each follow-up visit and the last visit can also have an event.

The time 't' for each follow up was generated using Cox- exponential model derived by Bender et.al (2005) [16] independently for each visit within each participant, without an upper limit on the follow up time. The survival time 't' for Cox-exponential model is derived as:

$$t_{ij} = -\frac{\log(u)}{\lambda \exp(\beta' x)}, \ u \sim U(0, 1)$$

The total duration for each participant ($t_i$) is the sum of all follow up visit times $\sum_{j=1}^{k} t_{ij}$, where k is the number of visits per participants ranging from 3 to 9. The random assignment of k is described later. Weights were generated using the follow up time between the last visit (k) from previous visit (k-1) and the expected duration from previous visit (k-1) until minimum expected total duration ($T_e$) for participation in the study. The minimum expected total duration ($T_e$) for the data is fixed as 20th percentile value of the total time simulated for all individuals. The weights are calculated for all simulated participants whose total duration is less than this minimum expected total duration ($T_e$).

The weight, denoted as $Weight_{TT}$, is calculated as the inverse of the ratio of time in the last follow up visit time to expected time in last follow up visit. Mathematically, it is defined as:

$$Weight_{TT} = \frac{1}{\frac{t_{ik}}{T_e - \sum_{j=1}^{k-1} t_{ij}}}$$

Where, $t_{ik}$ is the follow up time between the $(k-1)^{th}$ visit and the $k^{th}$ visit for $i^{th}$ participant, $T_e$ is the minimum expected total duration and $\sum_{j=1}^{k-1} t_{ij}$ is the total time duration till k-1 visit. For example: a participant who has already contributed 100 days to a study and whose last visit interval should have been 14 days but truncated at 7 days has weight 2. The schematic representation of the weighting mechanism is shown in Fig 1.

This weight is similar to sampling weights in the survey design. To limit the very high values of weights that were computed for very small simulated total time simulated randomly from the parametric model, a constant of 20 was added to each simulated follow up time $t_{ij}$.

$$t_{ij} = -\frac{\log(u)}{\lambda \exp(\beta' x)} + 20, \ u \sim U(0, 1)$$

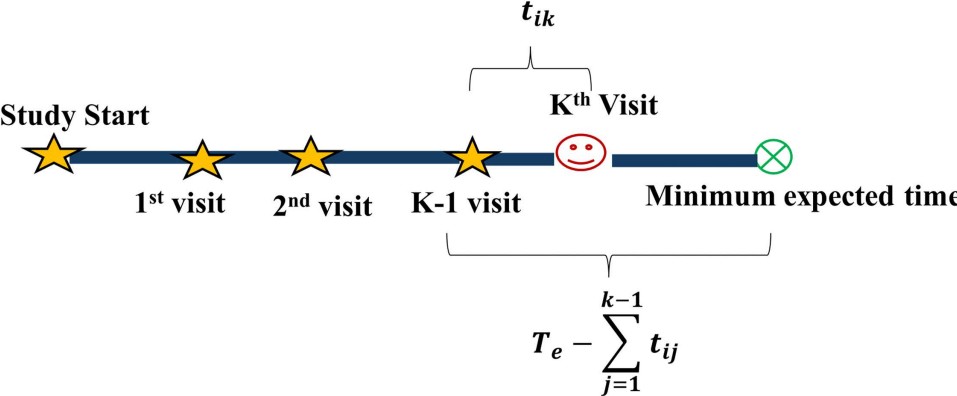

**Fig 1. Schema for weighting mechanism.**

Weights higher than 95th percentile [17] were truncated to avoid data sparseness. The constant value of 20 days was chosen to reflect a reasonable minimum follow up interval between hospital visits by pregnant women based on whom the simulation was designed (The real data detailed below provides more detail). Truncating extreme weights eliminated outliers in follow up duration and improved the stability of model estimates.

The model parameter u is generated using the standard uniform distribution U(0,1), for the Cox exponential model, where $\beta = 3$ and $\lambda = 1$. Variability in the total contributed time in the study was achieved by varying the total number of follow up visits. We considered four different scenarios with high variability to low variability in total number of follow up visits between participants. The four scenarios had 3–9 visits, 4–9 visits, 5–9 and 6–9 visits respectively across participants (Table 1). We fixed that varying number of visits ranging from three to six were generated in 50% of the participants and the remaining 50% had at least seven visits or higher in all scenarios.

The data were simulated with 1000 samples from 1000 iteration with fixed model parameters for each simulation. Then, the PWP-GT and PWP-GT weighted models were applied to each simulated dataset, and beta estimates of a binary risk factor was estimated. The bias was estimated as the difference between the estimated coefficient and the true beta ($\beta = 3$) used for the simulation. The percent bias (PBIAS) quantifies the difference between the estimated parameter and its true beta as a percentage.

**Table 1. Distribution of total number of visits and proportion of individual allocated in each visits in the simulated sample for four different scenarios.**

| Total no of visits | Proportion of participants in each visit | | | |
|---|---|---|---|---|
| | Scenario 1 (3–9 Visits) | Scenario 2 (4–9 Visits) | Scenario 3 (5–9 Visits) | Scenario 4 (6–9 Visits) |
| 3 | 10% | – | – | – |
| 4 | 10% | 10% | – | – |
| 5 | 20% | 20% | 20% | – |
| 6 | 10% | 10% | 10% | 20% |
| 7 | 10% | 20% | 20% | 30% |
| 8 | 20% | 20% | 30% | 30% |
| 9 | 20% | 20% | 20% | 20% |

## Real data

The weighted PWP GT was applied to real-time data extracted (n = 4000) from a double-blinded clinical trial. The trial was conducted on pregnant women to study the non-inferiority of low dose calcium to high-dose calcium on pregnancy outcomes. The trial protocol was approved by the Harvard T. H. Chan School of Public Health Institutional Review Board (Ref. No. IRB18−0157), the Institutional Ethics Committee St. John's Medical college & Hospital (IEC Study Ref. No. 197/2017), and the Indian Council of Medical Research (Ref. No. 2018−0380). The details and results of the study is published previously [18,19]. To use the data from the trial for demonstration purposes, we obtained a waiver of consent from the Institutional Ethics Committee, St. John's Medical College & Hospital (IEC Study Ref. No. 44/2021). The data was accessed on 01/04/2024. For this analysis, we considered nonadherence to the drug regime as an event of interest, as it was measured at every follow-up, making the outcome a recurrent event. Nonadherence was defined as the consumption of less than 90% of pills. In this study, the follow-up was planned at four weekly intervals and each woman in the study was expected to have at least four visits. Women who had a truncated period of participation in the study due to early terminal events such as delivery, abortion, miscarriage, and loss to follow-up will add bias to the hazard estimates, as the risk of event reduces (Fig 2).

The weights were estimated based on the average gestational age at delivery among the study participants, which was 39 weeks. Women who had completed the study before 39 weeks for various reasons were assigned weights. The weights were estimated:

$$Weight_{TT} = \frac{1}{\frac{t_{ik}}{T_e - \sum_{j=1}^{k-1} t_{ij}}}$$

where $t_{ik}$ is the number of days from the $(k-1)$ th follow-up visit to the $k$ th (delivery) visit, $T_e$ is the minimum expected number of days to reach 39 weeks of gestation, and $\sum_{j=1}^{k-1} t_{ij}$ is the total number of days contributed up to the $(k-1)$ th visit.

For example, consider a woman who delivers at 37 weeks, before the expected 39-week minimum. If her last visit occurred at 36 weeks (i.e., 252 days), then $t_{ik}$ is 7 days (from 36 to 37 weeks), $\sum_{j=1}^{k-1} t_{ij}$ is 252 days, and $T_e$ is 273 days (39 weeks). The weight for her delivery visit is calculated as:

$$Weight_{TT} = \frac{1}{\frac{7}{273-252}} = \frac{1}{\frac{7}{21}} = 3.03$$

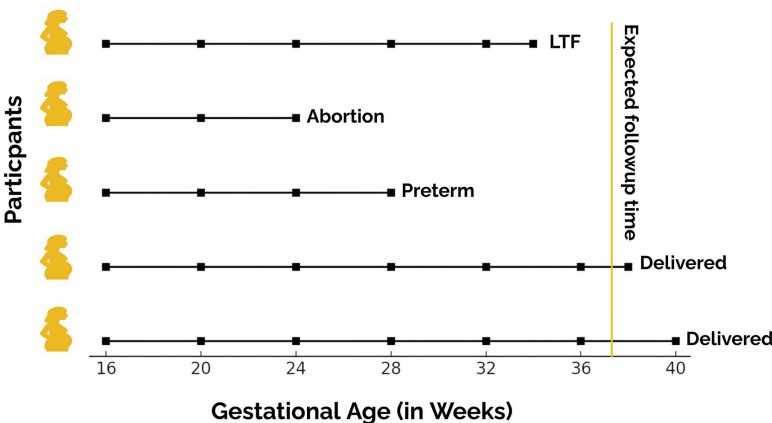

**Fig 2. Truncated follow-up patterns in pregnancy clinical trial.**

## Ethics approval

This is a statistical methodology study. The data used for application purpose was from a clinical trial which was approved by the Harvard T. H. Chan School of Public Health Institutional Review Board (Ref. No. IRB18−0157), the Institutional Ethics Committee St. John's Medical college & Hospital (IEC Study Ref. No. 197/2017), and the Indian Council of Medical Research (Ref. No. 2018−0380). The ethics for using data in the current study and waiver of consent was approved by Institutional Ethics Committee St. John's Medical college & Hospital (IEC Study Ref. No. 44/2021).

## Results

### Simulation

Table 2 shows that the Weighted PWP-GT model produced unbiased estimates across all four scenarios, with consistently lower percent bias. In Scenario 1, which had the highest variability in follow-up visits (3–9 visits), the weighted model had a percent bias of 1% compared to 1.3% in the unweighted model. Across all scenarios, the weighted estimates consistently showed lower absolute bias and percent bias (PBIAS) compared to the unweighted estimates (Table 2). The coverage probability, which estimates the proportion of times a confidence interval generated from repeated simulations contains the true parameter used in the simulation, remained above 94% across all four scenarios for both weighted and unweighted models.

### Real data

Data from 4000 pregnant women participants selected randomly were used for this demonstration. The mean age of participants was 23.80 ± 3.95, 65.6% had at least one drug non-adherence event during follow-up, with 35.3% experiencing a single event and 0.2% having six events. The total number of drug non-adherence visits was 4463 across 21,156 visits. 296 (7.4%) of the women reported family history of hypertension (Table 3).

Both unweighted and weighted Cox models indicate that a family history of hypertension is significantly associated with an increased hazard of recurrent non-adherence to medication. In the unweighted model, the hazard ratio (HR) is 1.23 (95% CI: 1.10–1.36) with a standard error (SE) of 0.065 and a p-value < 0.001. After applying inverse probability weighting, the HR decreases to 1.14 (95% CI: 1.03–1.25, p = 0.007) and the standard error is reduced from 0.065 to 0.054, reflecting an improvement in the precision of the estimate (Table 4).

**Table 2. Bias and mean beta estimates of total time weighted analysis.**

| Scenarios of maximum number of visits | True beta | Bias | Estimated beta | | Percent Bias (PBIAS) | Coverage Probability |
| --- | --- | --- | --- | --- | --- | --- |
| | | | Mean | Min, Max | | |
| **3 to 9 Visits** | | | | | | |
| Unweighted | 3 | 0.04 | 3.04 | 2.78, 3.38 | 1.3% | 94% |
| Weighted | 3 | 0.03 | 3.03 | 2.77, 3.37 | 1.0% | 94% |
| **4 to 9 visits** | | | | | | |
| Unweighted | 3 | 0.05 | 3.05 | 2.77, 3.40 | 1.6% | 95% |
| Weighted | 3 | 0.04 | 3.04 | 2.77, 3.41 | 1.3% | 94% |
| **5 to 9 visits** | | | | | | |
| Unweighted | 3 | 0.04 | 3.04 | 2.75, 3.43 | 1.3% | 94% |
| Weighted | 3 | 0.04 | 3.04 | 2.73, 3.41 | 1.3% | 94% |
| **6 to 9 visits** | | | | | | |
| Unweighted | 3 | 0.05 | 3.04 | 2.75, 3.40 | 1.3% | 95% |
| Weighted | 3 | 0.04 | 3.03 | 2.72, 3.39 | 1.0% | 95% |

**Table 3. Distribution of non-adherence visits and incidence rates among study participants (N = 4,000).**

|  | All women (N = 4000) |
|---|---|
| Age, Mean±SD | 23.80 ± 3.95 |
| Non-Adherence visits per women, n(%) |  |
| 0 | 1375 (34.4%) |
| 1 | 1410 (35.3%) |
| 2 | 780 (19.5%) |
| 3 | 291 (7.3%) |
| 4 | 108 (2.7%) |
| 5 | 28 (0.7%) |
| 6 | 8 (0.2%) |
| Total Non-Adherence Visits | 4463/21156 |
| Family History of hypertension |  |
| Yes | 296 (7.4%) |
| No | 3704 (92.6%) |

**Table 4. Weighted and unweighted model estimates from real time data.**

| Non-adherence | Haz. ratio (95% C.I) | Std. Err. | P-value |
|---|---|---|---|
| Family History of hypertension |  |  |  |
| Unweighted | 1.23 (1.10, 1.36) | 0.065 | <0.001 |
| Weighted | 1.14 (1.03, 1.25) | 0.054 | 0.007 |

## Discussion

This study developed and validated a simple time-based weighting approach for obtaining the unbiased hazard ratio for a covariate in a recurrent event model with fixed follow up intervals and varying total contribution time in clinical trials. Traditional survival models assume that all participants contribute equally to risk estimation. However, in real-world clinical trials, study dropout or early termination due to competing events often shorten follow-up durations, introducing systematic bias. Such truncation in time to recurrent event analysis may lead to a less precise and biased estimation in the risk factor analysis, as it reduces the risk of occurrence of the event. This study highlights the need for adjusting follow-up durations to ensure accurate risk factor estimation, particularly in clinical trials with recurrent event outcomes.

The findings from both simulation and real data analyses confirm that truncated follow-up can distort hazard estimates in recurrent event studies. In the simulated datasets, the weighted PWP-GT model consistently showed smaller percent bias and improved precision across all scenarios compared to the unweighted model. Similarly, in the real trial data, applying the time-based weights reduced the standard error and slightly attenuated the hazard ratio, indicating more accurate and precise estimates. These results demonstrate that accounting for follow-up variability can meaningfully improve the accuracy of risk estimation in recurrent event models.

Failure to adjust for truncated follow-up durations can substantially affect epidemiologic inference [14]. For example, in the real-time dataset used in this analysis from a clinical trial, the event non-adherence to the supplement is measured at a fixed follow-up interval of 4–6 weeks until delivery. As women could not complete the study and ended the study sooner than the expected days of follow-up, their risk of non-adherence reduced because the women were likely to be non-adherent to calcium supplement if they stayed longer in the study.

The sampling weight in time to event analysis was introduced to handle selection bias in case cohort design [20]. Later the weighting concepts evolved from sampling weights to inverse probability weights. Several studies applied inverse

probability weighting (IPW) through inverse probability of treatment weighting (IPTW) to address measured confounding in observational settings [8,9,13]. These weighting approaches allow for estimation of marginal treatment effects by creating a pseudo-population in which treatment is independent of baseline covariates. Additionally, inverse probability of censoring weighting (IPCW) was effectively used to handle informative censoring and selection bias, particularly in settings with a high risk of death or dropout [10,12]. In addition, approaches that account for competing risks [13] and incorporate survey design weights [11] highlight the relevance of choosing weighting strategies that align with the study design and data structure.

Although several weighting methods have been proposed in the literature, none were designed to correct the bias arising from truncated follow-up in clinical trials with recurrent events. This study provides an essential weighting scheme based on follow up time duration to adjust for the bias induced by truncation follow up duration in the clinical trial with recurrent events.

A key limitation of this study is that the current simulations and real-data application were conducted under conditions where other sources of bias, such as competing events or informative censoring were absent. Consequently, the findings primarily reflect the performance of the proposed method in settings where truncation is the dominant source of bias. Future work should examine the robustness and generalizability of the method in more complex scenarios that involve multiple, simultaneous assumption violation.

## Conclusion

The shorter or truncated duration of follow-up in a clinical trial with fixed intervals can result in fewer observed recurrent non-adherence events. This reduced event occurrence can introduce bias and reduce precision of the estimates from the Cox proportional hazards model (14). Our study demonstrates that applying time-based weighting effectively reduces this bias and improves precision through simulated and real time data. This novel methodology has significant implications for clinical trial design and epidemiologic studies, ensuring more reliable risk estimates in analyses of recurrent events.

## Supporting information

**S1 File. R codes.**
(DOCX)

## Acknowledgments

We thank **Dr. Wafaie W. Fawzi**, Department of Global Health and Population, Harvard T.H. Chan School of Public Health, Boston, Massachusetts, USA for allowing us to use the real time data from the trial. We thank **Dr Denis Xavier**, Professor & Head, Department of Pharmacology, St.John's Medical College, Bangalore, India, **Dr Srikant I Bangdiwala** – Director of Statistics, Population Health Research Institute, Hamilton, Canada, **Dr Shankar Viswanathan** – Vice Chair and Director of Quantitative Methods Core at the Department of Healthcare Delivery and Population Sciences at UMass Chan Medical School – Baystate, Springfield, Massachusetts, USA, and **Dr Binu VS** – Associate Professor, Department of Biostatistics, Dr. M.V. Govindaswamy Centre National Institute of Mental Health and Neuro Sciences, Bangalore, India, for their guidance and critical comments in the analysis as members of J.M.R. Doctoral advisory committee.

## Author contributions

**Conceptualization:** John Michael Raj A, Tinku Thomas.

**Data curation:** John Michael Raj A.

**Formal analysis:** John Michael Raj A.

**Methodology:** John Michael Raj A, Tinku Thomas.

**Resources:** Pratibha Dwarkanath.

**Software:** John Michael Raj A, Tinku Thomas.

**Supervision:** Tinku Thomas, Pratibha Dwarkanath.

**Writing – original draft:** John Michael Raj A.

**Writing – review & editing:** John Michael Raj A, Tinku Thomas, Pratibha Dwarkanath.

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
