## [Decision Letter · Decision Letter 0]

16 Oct 2025

Dear Dr. Thomas,

Thank you for submitting your manuscript to PLOS ONE. After careful consideration, we feel that it has merit but does not fully meet PLOS ONE’s publication criteria as it currently stands. Therefore, we invite you to submit a revised version of the manuscript that addresses the points raised during the review process.

We look forward to receiving your revised manuscript.

Kind regards,

Farshid Danesh, Ph.D.

Academic Editor

PLOS ONE

Reviewers' comments:

Reviewer's Responses to Questions

1. Is the manuscript technically sound, and do the data support the conclusions?

Reviewer #1: Partly

Reviewer #2: Partly

2. Has the statistical analysis been performed appropriately and rigorously?

Reviewer #1: Yes

Reviewer #2: Yes

3. Have the authors made all data underlying the findings in their manuscript fully available?

Reviewer #1: No

Reviewer #2: No

4. Is the manuscript presented in an intelligible fashion and written in standard English?

Reviewer #1: Yes

Reviewer #2: Yes

Reviewer #1: - I found this Manuscript to be an interesting and important contribution to the field of recurrent event analysis. The author focused on a real and often overlooked problem in clinical trials, truncated follow-up due to early termination or dropout. As they point out with the example from the calcium supplementation trial in pregnant women, women who delivered earlier or left the study seemed to have fewer non-adherence events. This could easily be misinterpreted as better adherence, when in fact it is simply a result of less time at risk. The proposed weighting approach is a clear and practical way to address this problem.

- The method itself is described clearly, with mathematical detail and worked examples. I appreciated the example where a participant who should have had 14 days of follow-up but only contributed 7 days receives a weight of 2. This helps the reader to see the logic behind the method. The simulations are also well designed. By setting up scenarios with different numbers of visits (3–9, 4–9), the author show that the weighted model consistently reduces bias compared to the unweighted version. The improvement is modest in percentage terms (for example, bias drops from 1.3% to 1.0%), but the consistency across scenarios supports the validity of the approach.

- The real trial data application strengthens the paper. In the unweighted model, a family history of hypertension gave a hazard ratio of 1.23 for non-adherence, while in the weighted model it dropped slightly to 1.14 with a smaller standard error. This illustrates the practical benefit of the method, slightly lower effect size but more precise estimates. That is exactly the kind of correction one would hope for when adjusting for truncated follow-up.

The authors mention other common methods, like IPCW (inverse probability of censoring weighting), frailty models, and joint models, but they never actually show how their new method compares to these. It would help readers if the authors did a side-by-side test, at least in the simulations, so we can see how well their approach performs compared to the others.

- In the simulation section, the authors say they added a constant of 20 days to avoid very large weights, but they don’t explain clearly why this number was chosen or whether different choices would change the results. Similarly, they cut off (or truncated) the weights above the 95%, which seems like a reasonable step, but again there is no explanation of why that cutoff was used. A little more detail here would make the methods easier to understand and give readers more confidence in the choices that were made.

Reviewer #2: Partially. The idea is sensible and the simulation + applied example provide preliminary evidence that the proposed time-based weighting reduces bias and can improve precision in some settings. However, several methodological choices and omissions limit how strongly the current results support a blanket conclusion that the weighting “effectively addresses” truncation bias across realistic trial settings.

Partially. The overall analytic strategy is reasonable (PWP-GT + time-based weights), but several omissions and methodological details must be added and some additional analyses performed to demonstrate the method’s robustness.

No — the current Data Availability Statement does not meet PLOS ONE policy. The manuscript states that the data “cannot be shared publicly because it is part of a clinical trial dataset. Data are available from the Principal investigator for researchers who meet the criteria for access to confidential data.” This is explicitly not sufficient under PLOS policy unless a compelling, specific exception is justified and documented.

Mostly intelligible but needs editing. The manuscript is generally readable; the structure is logical. However, there are multiple typographical and grammatical errors, slightly awkward phrasing, and places where clarity can be improved.

Do you want your identity to be public for this peer review? For information about this choice, including consent withdrawal, please see our Privacy Policy

Reviewer #1: No

Reviewer #2: No

---

## [Author Response · Author response to Decision Letter 1]

18 Nov 2025

Academic Editor:

Comment:

The important point of this article is the failure to follow the standard structure of academic writing based on the journal's author's guide. In other words, the standard structure includes the following: Introduction, Literature Review, Main Objective and Questions, Methodology, Findings, Discussion and Conclusion, Limitations and Suggestions for Future Research, and References. The present article should be restructured based on the stated format. It is also necessary to write a conclusion and inference from the literature review at the end of this section. The findings section of the article should be answered in the order of the research questions and the findings should be reported in the order of the questions.

Response:

Thank you very much for your time and for the detailed feedback on my manuscript.

I would like to respectfully clarify that the structure of the submitted article fully adheres to the PLOS ONE author guidelines provided on the journal’s official website (https://journals.plos.org/plosone/s/submission-guidelines

As per these guidelines, the manuscript was organized into the required sections: Title Page, Abstract, Introduction, Materials and Methods, Results, Discussion, and Conclusion, followed by Acknowledgments and References. This structure aligns with the “Beginning,” “Middle,” and “Ending” sections described by the journal, and was followed exactly to ensure compliance with their formatting standards.

The conclusion is based on the available literature and the findings from our study. We have checked the manuscript and the results are presented in the order of the objective and methods.

I sincerely appreciate your review and guidance; the present structure is consistent with PLOS ONE’s published author guidelines.

Reviewer #1: -

Comments:

I found this Manuscript to be an interesting and important contribution to the field of recurrent event analysis. The author focused on a real and often overlooked problem in clinical trials, truncated follow-up due to early termination or dropout. As they point out with the example from the calcium supplementation trial in pregnant women, women who delivered earlier or left the study seemed to have fewer non-adherence events. This could easily be misinterpreted as better adherence, when in fact it is simply a result of less time at risk. The proposed weighting approach is a clear and practical way to address this problem.

The method itself is described clearly, with mathematical detail and worked examples. I appreciated the example where a participant who should have had 14 days of follow-up but only contributed 7 days receives a weight of 2. This helps the reader to see the logic behind the method. The simulations are also well designed. By setting up scenarios with different numbers of visits (3-9, 4-9), the author show that the weighted model consistently reduces bias compared to the unweighted version. The improvement is modest in percentage terms (for example, bias drops from 1.3% to 1.0%), but the consistency across scenarios supports the validity of the approach.

- The real trial data application strengthens the paper. In the unweighted model, a family history of hypertension gave a hazard ratio of 1.23 for non-adherence, while in the weighted model it dropped slightly to 1.14 with a smaller standard error. This illustrates the practical benefit of the method, slightly lower effect size but more precise estimates. That is exactly the kind of correction one would hope for when adjusting for truncated follow-up.

Response:

We sincerely appreciate the reviewer’s thoughtful and encouraging comments. We are grateful that you recognized the importance of addressing bias caused by truncated follow-up in recurrent event analysis and your positive evaluation of the proposed weighting approach.

We are pleased to know that the mathematical description, worked example, and simulation design made the method’s logic and implementation clear. Your observation that the bias consistently decreases, even if modestly, perfectly reflects our objective of developing a practical and statistically sound adjustment that improves estimation while keeping the model implementation straightforward.

We also thank you for highlighting the real trial application, where the weighted model produced slightly attenuated but more precise estimates. We fully agree that this demonstrates the intended correction for differences in exposure time resulting from truncated follow-up.

We are truly grateful for your positive feedback and for recognizing both the methodological rigor and practical relevance of this work.

Comment:

The authors mention other common methods, like IPCW (inverse probability of censoring weighting), frailty models, and joint models, but they never actually show how their new method compares to these. It would help readers if the authors did a side-by-side test, at least in the simulations, so we can see how well their approach performs compared to the others.

Response:

We sincerely thank the reviewer for this valuable suggestion. We agree that comparing the proposed method with other existing approaches can often provide additional insights. However, the existing weighting methods are not meant to address the problem of truncated follow up and therefore in the present study, the proposed method was specifically developed to address bias arising from truncated follow-up. This is conceptually distinct from the issues handled by methods such as IPCW, frailty models, or joint models. We listed to existing weighting methods to highlight the gap and the need for this study. For this reason, we focused our simulations on evaluating the performance of the proposed approach under varying levels of truncation, which represents the specific gap our method aims to address. We have modified statements in the discussion section (line no 253 – 254) to make it very clear.

Comment:

- In the simulation section, the authors say they added a constant of 20 days to avoid very large weights, but they don't explain clearly why this number was chosen or whether different choices would change the results. Similarly, they cut off (or truncated) the weights above the 95%, which seems like a reasonable step, but again there is no explanation of why that cutoff was used. A little more detail here would make the methods easier to understand and give readers more confidence in the choices that were made.

Response:

We thank the reviewer for this thoughtful comment and for the opportunity to clarify these points. The constant value of 20 days was added to mimic the real trial data structure, where the minimum follow-up time between consecutive visits is 4 weeks. This constant helps to maintain realistic spacing between events and prevents simulations with unrealistic very short follow-up durations leading to generation of excessively large weights.

Regarding the truncation of weights at the 95th percentile, this choice follows the recommendation by Daniela Dunkler et al. (2018) in their paper “Weighted Cox Regression Using the R Package coxphw” (Journal of Statistical Software, Volume 84, Issue 2, (DOI: 10.18637/jss.v084.i02). Truncating extreme weights helps to reduce the influence of outliers and improve the stability of model estimates, which is particularly relevant when the distribution of weights is highly skewed.

These details have been added in the manuscript (Lines 127 to 131).

Reviewer #2:

Comment:

Partially. The idea is sensible and the simulation + applied example provide preliminary evidence that the proposed time-based weighting reduces bias and can improve precision in some settings. However, several methodological choices and omissions limit how strongly the current results support a blanket conclusion that the weighting "effectively addresses" truncation bias across realistic trial settings.

Partially. The overall analytic strategy is reasonable (PWP-GT + time-based weights), but several omissions and methodological details must be added and some additional analyses performed to demonstrate the method's robustness.

Response:

We thank the reviewer for this insightful comment and agree that the current analyses represent an initial step toward evaluating the proposed time-based weighting approach. In response, we have now explicitly acknowledged this point in the Discussion section. Specifically, we have added the following statement to highlight the study’s scope and limitations (line no: 257-262):

“A key limitation of this study is that the current simulations and real-data application were conducted under conditions that did not incorporate other potential sources of bias, such as competing events or informative censoring. Consequently, the findings primarily reflect the performance of the proposed method in settings where truncation is the dominant source of bias. Future work should examine the robustness and generalizability of the method in more complex scenarios that involve multiple, simultaneous assumption violations.”

This addition clarifies that our findings should be interpreted within the context of the current study design and that further work is needed to assess the method’s robustness under broader, more realistic conditions.

Comment:

No — the current Data Availability Statement does not meet PLOS ONE policy. The manuscript states that the data "cannot be shared publicly because it is part of a clinical trial dataset. Data are available from the Principal investigator for researchers who meet the criteria for access to confidential data." This is explicitly not sufficient under PLOS policy unless a compelling, specific exception is justified and documented.

Response: We have now added the codes to simulate the data used in the study in Supplementary material 1. In addition, the process of obtaining de-identified data used as real data in the paper is updated and detailed below.

The dataset used for demonstration in this manuscript is derived from a third-party randomized controlled trial: Dwarkanath P, Muhihi A, Sudfeld CR, Wylie BJ, Wang M, Perumal N, et al. Two Randomized Trials of Low-Dose Calcium Supplementation in Pregnancy. N Engl J Med. 2024;390(2):143–53. The authors do not own the rights to share these data; all data-sharing rights remain exclusively with the study Principal Investigator (PI).

As detailed in the original publication’s data-sharing policy, complete de-identified patient-level data from the trial may be made available to qualified researchers. Access is granted only upon formal request to the PI, accompanied by a research proposal and the necessary approvals from the Institutional Ethics Committee (IEC) and other relevant regulatory authorities.

The PI details, as provided in the original trial’s data-sharing statement, are:

Dr. Wafaie Fawzi

Harvard T.H. Chan School of Public Health

665 Huntington Avenue

Boston, MA 02115

Email: mina@hsph.harvard.edu

Comment:

Mostly intelligible but needs editing. The manuscript is generally readable; the structure is logical. However, there are multiple typographical and grammatical errors, slightly awkward phrasing, and places where clarity can be improved.

Response:

We thank the reviewer for the constructive comment regarding the manuscript’s language and clarity. In response, we have thoroughly revised the text to correct typographical and grammatical errors and to refine phrasing throughout. Several sentences have been shortened or restructured to enhance readability and ensure a clearer and more concise presentation of the material.

---

## [Decision Letter · Decision Letter 1]

14 Dec 2025

Adjusting for Truncated study duration in Recurrent Event Analysis: A Weighting Approach for Clinical Trials

PONE-D-25-35687R1

Dear Dr. Thomas,

We’re pleased to inform you that your manuscript has been judged scientifically suitable for publication and will be formally accepted for publication once it meets all outstanding technical requirements.

Kind regards,

Farshid Danesh, Ph.D.

Academic Editor

PLOS One

Additional Editor Comments (optional):

Reviewers' comments:

Reviewer's Responses to Questions

**Comments to the Author**

Reviewer #1: All comments have been addressed

Reviewer #2: All comments have been addressed

2. Is the manuscript technically sound, and do the data support the conclusions?

Reviewer #1: Yes

Reviewer #2: Yes

3. Has the statistical analysis been performed appropriately and rigorously?

Reviewer #1: Yes

Reviewer #2: Yes

4. Have the authors made all data underlying the findings in their manuscript fully available?

Reviewer #1: (No Response)

Reviewer #2: Yes

5. Is the manuscript presented in an intelligible fashion and written in standard English?

Reviewer #1: (No Response)

Reviewer #2: Yes

Reviewer #1: Thank you for the revised manuscript. I have reviewed the updated version, and the author have addressed all my comments. The weighting approach rationale, the explanation for the constant and truncation choices, and the clarification regarding comparisons with other methods are now clearly presented. The revisions improve the clarity, justification, and transparency of the methodology.

Reviewer #2: The authors have fully addressed the comments raised in the previous round. The manuscript now presents a technically sound and clearly described analysis. The simulations are well-designed, and the real-data example is appropriate and strengthens the work. The conclusions are drawn cautiously and are now aligned with the results. The Data Availability Statement has been revised to meet PLOS ONE policy: code for the simulations is provided, and access procedures for the third-party clinical trial data are clearly documented. In addition, the manuscript has been edited for clarity and readability, and no remaining issues affecting intelligibility were noted. I have no concerns regarding research or publication ethics. I recommend the manuscript for publication.

**Do you want your identity to be public for this peer review?** For information about this choice, including consent withdrawal, please see our Privacy Policy

Reviewer #1: No

Reviewer #2: No

---

## [Editor Report · Acceptance letter]

PONE-D-25-35687R1

PLOS One

Dear Dr. Thomas,

I'm pleased to inform you that your manuscript has been deemed suitable for publication in PLOS One. Congratulations! Your manuscript is now being handed over to our production team.

Kind regards,

on behalf of

Associate Professor Farshid Danesh

Academic Editor

PLOS One